# Flow Regime Recognition and Dynamic Characteristics Analysis of Air-Water Flow in Horizontal Channel under Nonlinear Oscillation Based on Multi-Scale Entropy

**DOI:** 10.3390/e21070667

**Published:** 2019-07-08

**Authors:** Bo Sun, He Chang, Yun-Long Zhou

**Affiliations:** Department of Energy and Power Engineering, Northeast Electric Power University, Jilin 132012, China

**Keywords:** gas-liquid two-phase flow, heaving motion, multi-scale entropy

## Abstract

Gas-liquid two-phase flow behavior in horizontal channel under heaving motion showed unique dynamic characteristics due to the complex nonlinear interaction. To further establish a description model and investigate the effects of heaving motion on horizontal gas-liquid flow, experiments in a wide range of vibration parameters and working conditions were carried out by combining vibration platform with two-phase flow loop. It was found that the flow regimes under heaving motion showed significant differences compared to the ones expected in steady state flow under the same working conditions. Increasing vibration parameters showed an obvious impact on fluctuation degree of gas-liquid interface by visualizing high-speed photographs. A method based on multi-scale entropy was applied to identify flow regimes and reveal the underlying dynamic characteristics by collecting signals of pressure-difference. The results indicated that the proposed method was effective to analyze gas-liquid two-phase flow transition in horizontal channel under heaving motion by incorporating information of flow condition and change rate of multi-scale entropy, which provided a reliable guide for flow pattern control design and safe operation of equipment. However, for slug-wave and boiling wave flow, an innovative method based on multi-scale marginal spectrum entropy showed more feasible for identification of transition boundary.

## 1. Introduction

For the sake of cutting carbon and other greenhouse gas emissions caused by continuing combustion of fossil fuels for supplying the growing demand electricity, nuclear power systems are extensively applied in electricity generation with advantages including economic benefits, safety and sustainability [1]. The most significant difference between barge-mounted and land-based nuclear equipment is the effect of different motions contributed by sea waves such as rolling, heaving motion and pitching [2,3], which means different variations of periods, amplitude and other parameters will lead to unsteady flow and process as mentioned by Pendyala [4]. Therefore, it has practical significance to conduct research on gas-liquid flow behavior in channel under heaving motion. Cao et al. [5] proposed a new empirical correlation equation for flow regime transition line in channel with round section. The results verified that heeling of channel will affect effective forces acting on fluid such as gravitation, pressure and friction, which led to a change in momentum, heat and mass transfer characteristics. Islam et al. [6] presented an efficient and robust calculation scheme for two-phase, one-dimensional steady state steam condensation in the presence of CO_2_, based on conservation rules and thermodynamic phase relations. Xing et al. [7,8] investigated frictional resistance of adiabatic two-phase flow under rolling motion conditions, indicating that fluctuation amplitudes of instantaneous frictional pressure gradient increased with the decrease of flow rate. A similar experimental study had been generated by Li et al. [9]. The agreements of conclusions confirmed that instantaneous frictional pressure drop has a strong relationship with rolling parameters. From aforementioned work, it is clear that flow pattern and characteristics under dynamic conditions will be significantly different from that under stable conditions.

As a time-variant non-liner dynamic system, the identification of flow patterns of gas-liquid two-phase flow has aroused extensive interest both at industrial and scientific levels. Many measuring techniques such as high-speed camera [10], differential pressure drop [11] and capacitance sensors [12] have been adopted in experiments to extract flow information for analyzing with method of imaging analysis, statistical analysis, time-frequency analysis of either wavelet transform or Hilbert transform, chaotic analysis and entropy analysis [13,14]. However, methods mentioned above usually use one independent measurement data series which will only provide flow information in either local or average manner. Owing to the complexity of gas-liquid two-phase flow, a method of multiple-point measurement is needed. 

As an index of complexity and regularity of systems, entropy has been used for data analysis of nonlinear systems. Since Pinus [15] proposed approximate entropy in 1991, it was widely used in physiological and medical signal processing. However, the calculation results have a close relation with the selection of parameters, showing inapplicability for data analysis with large quantities or noise. After that, an improved approximate entropy method was presented by Richman [16]. Sample entropy indicated more applicability in analyzing complex signals [17]. Meanwhile, entropy increases with the degree of disorder of a dynamic system and reaches its maximum for a completely random system, such as the two methods mentioned above. Nevertheless, the increase in entropy is not always explicitly related to a higher degree of overall complexity. Therefore, on the basis of sample entropy, Costa [18] put forward a multi-scale entropy theory which was applied to recognize vertical conductance fluctuation signals of two-phase flow pattern in rising pipes [19]. It was found that multi-scale entropy could quantify the interdependence between entropy and scale through evaluating sample entropy of univariate time series that coarse grained at multiple temporal scales, and had been successfully applied on analyzing the complexity of physiologic signals [20]. In the case of gas-liquid two-phase flow, it has been applied to flow pattern recognition. 

Zhou et al. [21] successfully applied multi-scale entropy theory to analyze differential pressure signals in a 3 × 3 bundle channel, showing a good robustness of revealing the complexity of a signal. Li et al. [22] carried out the symbolic dynamics information entropy extraction of the conductance signals of conventional gas-liquid two-phase flow and analyzed their dynamic typical flow pattern behavior characteristics. It was found that the change rate of small-scale sample entropy could distinguish different typical flow patterns by analyzing gas-liquid two-phase flow characterization with electrical resistance tomography and multivariate multi-scale entropy analysis [23,24]. With the continuous improvement of signal theory, Li et al. [25] first used digital information to characterize and distinguish stochastic from chaotic process, and they noted that it is important to use multiple scale when measuring the different states of a dynamic system. Considering the characteristics of Henon and Lorenz systems, it was shown by Zhu et al. [26] that the system with strong autocorrelation is sensitive to that change of scale which verifies that the Henon system can well describe dynamic characteristics of complex nonlinear system better than Lorenz system. 

Although multi-scale entropy analysis has made great progress in characterizing physiological and biological complex signals at multi-space-time scales, most of its analysis methods are based on characteristics of multi-scale entropy distribution pattern to qualitatively identify complex research objects. Thus, there are still some deficiencies in extraction of invariant characteristics exhibited by multi-scale entropy when changing with scale. As one of the most mature signal testing technologies [27], differential pressure signal was used in present study to analyze the dynamic characteristics of gas-liquid two-phase flow pattern in channel under heaving motion. Firstly, this paper compared the distribution of gas-liquid two-phase in channel under stable condition and heaving motion, which illustrated that the influence of vibration motion on fluid flow in horizontal channel. Based on that, the differential pressure signals were collected and analyzed by wavelet denoising. Furthermore, based on change rate of multi-scale entropy and varying characteristics of that at different scales, it was analyzed comprehensively on a multi-scale entropy of differential pressure drop fluctuation of gas-liquid two-phase flow in channels under heaving motion. Considering that the magnitude of multi-scale entropy may have a relationship with vibration parameters, this work also investigated the variation tendency of multi-scale entropy with the increasing vibration amplitude and frequency. Based on this reasoning, the method adopted in this paper can not only revealed the inherent flow mechanics of gas-liquid two-phase flow, but is also available for fluid flow in channels under dynamic conditions.

## 2. Experimental Setup 

Gas-liquid two-phase flow behavior in horizontal channel under heaving motion was studied experimentally. The experimental apparatus was composed of vibration platform, gas-liquid two-phase flow loop and data acquisition system. Detailed illustrations will be given as follows.

### 2.1. Vibration Platform

As illustrated in Figure 1, the vibration platform is mainly composed of an industrial control computer, multi-channel parallel collection, asynchronous collaborative control software, capacitance transducer, digital switching power amplifier and electromagnetism oscillator. Displacement, frequency and acceleration data were obtained through multi-channel parallel collection of industrial control computer. After decomposition using asynchronous cooperative control software, unique control function modules of the vibration data were transformed through capacitance transducer into voltage signals. Rated parameters of one vibration platform are shown in Table 1.

Taking advantage of MOSFET’s high frequency switching capabilities, the digital switching power amplifier amplifies voltage signals by digital circuit and outputs them to four separately installed electromagnetic oscillators which can restore the signals of displacement, frequency and acceleration, respectively. At the beginning of the experiment, in order to minimize error, it is necessary to debug vibration platform for ensuring the synchronization of vibration between test section and vibration platform. The instantaneous vibration displacement, velocity and acceleration can be approximated by
(1)$$Z=A\sin\left(\omega t\right)=A\sin\left(2\pi ft\right)$$
(2)$$V=\frac{dZ}{dt}=2A\pi f\cos\left(2\pi ft\right)$$
(3)$$a=\frac{dV}{dt}=-4A{\left(\pi f\right)}^{2}\sin\left(2\pi ft\right)$$
where *A* is vibration amplitude, *f* is vibration frequency, and *t* is time.

### 2.2. Experimental Loop

As shown in Figure 2, filtered and deionized water was used as liquid phase in the fluid flow system. The liquid phase was circulated from a tank by a centrifugal pump with flow rate ranging from 0 to 5 m^3^/h, controlled and measured by an electromagnetic flow meter and needle valves. Gas phase was delivered by air compressor with a flow rate of 0.1 to 35 m^3^/h, which was adjusted by changing the opening of throttle valve and recorded by an orifice plate flow meter. To diminish the impact of nonlinear oscillations on fluid flow system, rubber tubes were placed closely on the test section. After passing a through phase mixer, working fluid flowed in a horizontal channel of length equal to 2 m.

The test section consists of a circular channel made of plexiglass with an inner diameter of 35 mm. The test section was fixed to the vibration platform, which was moved using an electromagnetic vibration system. The movement was within the maximum vibration parameters controlled by the vibration control system. The static state of the vibration platform was considered as equilibrium situation of experiment. After passing through the test section, water flowed back to the water tank via another rubber tube for recycling. A bypass valve was used to recycle excess water which was pumped back into the water tank. A needle valve was provided to control the flow to test section and a gate valve was used to control bypass. All outlet parts of backwater pipes were immersed in water and hence the water tank can be considered as a pressurizer with atmosphere pressure. The experimental method conducted in the present study was the same as that used in our previous researches on gas-liquid two-phase flow in channel under stable state [28], which was adjusting liquid flow rate to a certain experimental value and keeping it constant at first, and then slowly increasing gas flow rate with the experiment going forward. After completing experimental content at this condition, adjusting liquid flow rate and repeating the above steps is continued until the work is done. A differential pressure transducer was employed to record the two pressure measure points which have a distance of 1400 mm dynamically. To ensure the flow was fully developed, one is fixed 400 mm distance from entrance and the other is 1800 mm. During the experiment, identification of fluid flow phenomena for each condition was based on photos captured by high speed camera which has a capture rate of 512 frames per second. Besides that, a computer-based data acquisition system was also used to record fluid flow rate and pressure drop. 

### 2.3. Instrumentations

The electromagnetic flow meter has an overall measurement uncertainty of ±0.3%. An orifice flow meter has a measurement uncertainty of ±0.5%. The differential pressure transducer (range 0~2 bar) has an overall uncertainty of ±0.2%. The validations of these measuring instruments are performed by conducting experiments under stable conditions. Measurement uncertainties of channel ID and length are less than ±0.03 mm. During each experiment, a computerized data acquisition with analog-to-digital (A/D) cards is employed to record data under a sampling rate of 512 samples per second, which is sufficient to recognize flow regime and track variation of parameters. At the same time, the test for dynamic response time of instruments employed in experiments is needed for ensuring the accuracy of experiments, which is no more than 100 ms each time. The operating temperature used for our experiments is ambient temperature.

### 2.4. Experimental Results

Various high-speed photos of flow pattern under heaving motion taken for comparison with stable condition were shown in Figure 3. A newly developed flow regime map in horizontal channel (35 mm) under stable conditions is illustrated in Figure 4a. This regime map is based on the definitions of each flow regime along with flow regime boundaries suggested by Ran et al. [29] and Mandhane [30]. In the figure, fine dashed lines represent previous conclusions proposed by Ran et al. [29], and dotted lines represent suggestions by Mandhane [30]. Beyond that, a flow regime transition boundary map of fluid flow in horizontal channel under heaving motion is also shown in Figure 4b. 

Based on the test conditions investigated above, it can be seen that heaving oscillation causes gas-liquid two-phase flow behavior to be more complicated in horizontal channel. As a result, there is a higher chance for fluid flow to be affected by different oscillation parameters. Considering that vibration parameters are composed of vibration frequency and amplitude, reliable experimental databases in different vibration parameters were established.

The videos were captured by high-speed camera with different resolutions depending on flow regime. Effect of vibration parameters on flow regime are shown in Figure 5. 

As shown in Figure 5, it was found that the intensity of the collision between fluid and channel wall was increased as vibration frequency or amplitude increased. It is believed that when relevant vibration parameters increase, the circumferential turbulence in liquid phase increases at the same flow condition. Correspondingly, bubbles are more concentrated near wall surface and will form large elongated gas column. Similarly, the increase in vibration amplitude can also be regarded as an increase in the intensity of nonlinear oscillation. Consequently, it has similar effect on gas-liquid interface with vibration frequency.

From the current experiment, although vibration parameters have a certain effect on fluid flow characteristics, they will only affect the interface fluctuation intensity, instead of flow pattern transition boundary. To verify the assumption, based on the results of flow regime identification, newly developed transition boundaries in horizontal channel under heaving oscillation with various vibration parameters are shown in Figure 6.

Figure 6 summarized flow regimes found in each scenario: It can be found that trend distribution of the whole flow pattern has an outward expansion tendency with the slug flow as the center, which can be contributed to impact of nonlinear oscillation. Due to the effect of oscillation, gas phase is more likely to form small bubbles in channel. As a result, bubbles are more concentrated near the top wall of the channel, leading to a higher chance for bubbles to collapse or form large elongated gas columns. As one of the most favorable flow patterns for heat transfer, slug flow is widely used in engineering because of its good heat and mass transfer performance. This, in turn, suggests that nonlinear oscillation can enhance heat transfer. The reasons are inferred as follows:

On one hand, vibration of channel is equivalent to applying an additional velocity to fluid in near-wall region along the direction of oscillation. By synthesizing velocity of additional and original speed, it will disturb formation and development of flow boundary layer near channel wall. Accordingly, it will lead to perturbation in thermal boundary layer by reducing the thickness of thermal boundary. Therefore, nonlinear oscillation can reduce heat transfer resistance. 

On the other hand, due to the input of vibration energy, energy of flow field itself increases rapidly which is finally converted by real work, indicating that vibration has effect of work on flow field.

Beyond that, it should be noted that vibration parameters have insignificant effect on flow pattern transition boundary, which also confirms the above statement. In other words, the same method, based on multi-scale entropy, can be applied to identify flow patterns under various vibration parameters. Details will be illustrated in Section 3.3.3.

## 3. Analysis of Flow Pattern Based on Multi-Scale Entropy

### 3.1. Theory Basis

Sample Entropy (SampEn) was proposed by Richman and Moonman [14] in 2000, which was of practical value in evaluating the complexity of physiological time series [29] and analyzing dynamic couplings over multiple scales of nonlinear multichannel data [30]. Compared with Approximate Entropy (AE) which is also a nonlinear dynamic parameter used to quantify the regularity and unpredictability of time series fluctuation, the result of SampEn has a better relative uniformity. SampEn is calculated in the following steps:

I. Define a one-dimensional discrete time scale series *U* of length *N* taken at regular intervals: {*u*(*j*): *j* = 1, 2, …, *N*}, and a given embedding dimension *m*, a sequence of reconstruction time series obtained from *u*(*j*), defined as *X_m_*(*i*) where { *i*|1 ≤ *i* ≤ *N* − *m* + 1}, can be illustrated as *X_m_*(*i*) = {*u* (*i* + *k*), 0 ≤ *k* ≤ *m* − 1}. The distance between two vectors can be obtained by calculating the Chebyshev distance, where the maximum distance between scalars corresponding to two vectors along any coordinate dimension can be expressed mathematically as
(4)$$d\left|X\left(i\right),X\left(j\right)\right|=\max\left\{\left|u\left(i+k\right)-u\left(j+k\right)\right|:0\leq k\leq m-1\right\}$$


II. The probability of vector *X_m_*(*i*) being similar to *X_m_*(*j*) is computed as (*N* − *m* − 1)^−1^ times the number of vectors *X_m_*(*j*), where 1 ≤ *j* ≤ *N* − *m*, within a similarity tolerance of *X_m_*(*i*) excluded self-matches can be presented mathematically as
(5)$$B_{i}^{m}\left(r\right)={\left(N-m-1\right)}^{-1}\sum_{j=1}^{N-m}H\left[d\left(x_{m}(i),x_{m}(j)\right)\right],i\neq j$$
(6)$$H\left[d\left(x_{m}(i),x_{m}(j)\right)\right]=\left\{\begin{array}{l} 1:d\left(x_{m}(i),x_{m}(j)\right)\leq r\\ 0:d\left(x_{m}(i),x_{m}(j)\right)\geq r \end{array}\right.$$
where *r* is a real positive value for the similarity tolerance, and *H* is Heaviside function.

III. Define the probability of vector *X_m_*(*i*) and *X_m_*(*j*) having Chebyshev distance within *r* as follows
(7)$$B^{m}\left(r\right)={\left(N-m\right)}^{-1}\sum_{i=1}^{N-m}B_{i}^{m}\left(r\right)$$


Similarly, *A_m_*(*r*) is defined as (*N* − *m* − 1)^−1^ times the number of vectors *X_m_*(*i* + 1) within a similarity tolerance of *X_m_*(*j* + 1), where 1 ≤ *j* ≤ *N* − *m*, *i* ≠ *j*, is shown as
(8)$$A^{m}\left(r\right)={\left(N-m\right)}^{-1}\sum_{i=1}^{N-m}A_{i}^{m}\left(r\right)$$


IV. Define SampEn as
(9)$$\mathrm{ampEn}\left(m,r\right)=\lim_{N\to \infty }\left\{-\ln\left|A^{m}\left(r\right)/B^{m}\left(r\right)\right|\right\}$$


After statistical analysis, SamEn can be calculates as
(10)$$\mathrm{SampEn}\left(m,r,N\right)=-\ln\left|A^{m}\left(r\right)/B^{m}\left(r\right)\right|$$


To express a higher degree of overall complexity of multi-scale time series, it is significant to incorporate multi-scale into design measures. Therefore, on the basis of SampEn, Costa et al. [18] proposed multi-scale entropy theory which quantifies the interdependence of entropy and scale by evaluating SampEn of univariate time series. The coarse grained at multiple temporal scales also applied the theory on analyzing complexity of physiologic signals. This method has illustrated a better representation on showing differences.

The time domain multi-scale coarse grained schemes are presented as follows:

I. Define a one-dimensional discrete time series as {*u*(*i*):*i* = 1, 2, …, *N*} where *N* is the number of samples in every channel.

II. Construct temporal scales by coarsely graining time series: With the set scale of 1, the reconstruction is the original time series. For a *τ*-scale, the time-series is coarse grained as {*y^τ^*(*j*) = 1, 2, …, *N*/*τ*} where *y^τ^*(*j*) can be obtained as
(11)$$y^{\tau }\left(j\right)=\frac{1}{\tau }\sum_{\left(j-1\right)\tau +1}^{j\tau }u\left(i\right),1\leq j\leq N/\tau$$


### 3.2. Multi-Scale Entropy of Typical Signals

To verify the availability of multi-scale entropy on characterizing different nonlinear signals of dynamic systems, typical and classic nonlinear signals were employed as examples including Logistic, Henon, Lorenz and sine chaotic time series. These several group of time series with different parameters were generated under the following condition.

1. Logistic time series
(12)$$x_{n+1}=ax_{n}\left(1-x_{n}\right),x_{n}\in \left[0,1\right],a\in \left[0,4\right],n=0,1$$


Set the initial condition *x*_0_ = 0.4, the coefficient *a* = 3.9.

2. Henon system time series (Henon and Heile [31])
(13)$$\left\{\begin{array}{l} x_{n+1}=1-\alpha {x_{n}}^{2}+y_{n}\\ y_{n+1}=\beta x_{n} \end{array}\right.$$
with the initial condition where *α* = 1.4, *β* = 0.3, *x*_0_ = 0, *y*_0_ = 0. The signal simulation programmed with MATLAB is illustrated in Figure 7.

3. Lorenz system time series (Zheng and Jin [32])
(14)$$\begin{array}{l} \frac{d_{x}}{d_{t}}=-\sigma x+\sigma y\\ \frac{dy}{d_{t}}=rx-y-xz\\ \frac{dz}{d_{t}}=-bz+xy \end{array}$$
where *σ* = 10, *r* = 28, *b* = 8/3. The signals produced with initial value where *x* = 2, *y* = 2, *z* = 20 was shown in Figure 8 by using iteration method of fourth-order Runde-Kutta.

4. Sine signal
(15)y=3sinx


The signal was presented in Figure 9 with the sampling interval as *π*/500.

Due to the noise characteristics of signals collected from the experiment (which is inevitable), the first three simulation variable of typical signals mentioned above were added with noise to obtain a signal-to-noise ratio of 30 dB. The sine plus noise is given by *y* = 3sin*x* + *p*·*y*_1_, where *y*_1_ is white Gaussian noise (wGn) sequence, and p is a random ratio parameter defined as 0.2 in present study. The distribution of multi-scale entropy of 4 typical system time series at different scales was shown in Figure 10, which can reflect the sensitivity of different typical signal dynamics to scale. The lengths of original time series of four signals are all 105. In Figure 8, scales from 1 to 20 were selected with scale 20 representing the worst correlation of each element in the selected range [33]. Scale 1 is the original time series, which maintains original correlation between system elements.

As shown in Figure 10, multi-scale entropy of Henon and Logistic signal have an overall similarity in variation tendency, indicating a rising trend in the first four scales while decreasing gradually until smooth and steady. The discrepancy between them from scale 3 to 6 can be revealed obviously by method of multi-scale entropy compared with SampEn method which is unable to distinguish Logistic and Henon time series due to very similar entropy of original time series, corroborating the advantages of multi-scale entropy in analyzing dynamic characteristics of complex nonlinear system. Differing from Henon and Logistic, the multi-scale entropy of Lorenz displays a random movement showing more complexity, however, the randomness of entropy produced by Lorenz is not that strong, showing partial linearization which rises gently in the first five scales and remains nearly constant until scale of 10, then increases rapidly till scale of 18, before finally declining in volatility. Considering entropy characteristics of the above three time series, multi-scale entropy method can well describe the definiteness of chaotic time series. Meanwhile, it can be found that multi-scale entropy of sine time series is regular and even keeping constant which is appeared after scale of 8, which is consistent with the periodicity and regularity of sine signal. With the input of noise, it can be noted that the algorithm is still feasible, showing an intensified anti-noising ability of algorithm as increasing scale. 

In order to verify the sensitivity of multi-scale entropy algorithm to data length, the characteristic of multi-scale entropy with four data lengths in 20 scales of slug flow and Gaussian noise were calculated, and are illustrated in Figure 11. 

As can be seen from Figure 11, there is little effect of data length on multi-scale entropy with specified condition. At scale of 12 in Figure 11a, all multi-scale entropy of four data lengths fluctuated downward and increased steadily until the scale of 18. From scale of 17 to 20 in Figure 11b, multi-scale entropy of all three data lengths tended to fluctuate except for *N* = 15,000 which shows a very minor discrepancy. Therefore, the coincident tendency of multi-scale entropy with different sequence lengths fully testified the robustness of the mentioned algorithm, which is of significance to improve operation speed. Besides that, although it is possible to shorten data length, it should be guaranteed that the selection of that can fully reflect the evolution of original signal.

### 3.3. Analysis of Flow Characteristics

#### 3.3.1. Frequency Analysis of Differential Pressure Signal

The differential pressure signals of typical flow regime showed in Figure 3 are presented in Figure 12, where the sampling frequency is 500 Hz, 5000 data points were obtained of gas-liquid two-phase flow under heaving motion with vibration frequency of 5 Hz and amplitude of 5 mm. Compared with differential pressure signal of gas-liquid two-phase flow in stable channel under the same working condition, the differential pressure signal of gas-liquid two-phase flow under dynamic conditions has not changed much except for a faster varying frequency, whereas the signals of each typical flow regime are significantly different. 

As can be seen from Figure 12, differential pressure drops present an irregular sinusoidal behavior which is probably caused by lateral velocity of near-wall fluid and the interaction between gas and liquid phase, illustrating that the whole channel pressure is governed by fluid flow rate fluctuation due to heaving motion. For bubbly flow as shown in Figure 12a, differential pressure drop presents the most irregular and violent variation trend, which indicates that the signal has higher complexity and more information. Figure 12b denotes a relatively sharp and high frequency fluctuation of bead flow which is also a novel flow regime compared with flow conditions in stable channel, indicating that the signals have characteristics of relatively high complexity and more information. Compared with bead flow, differential pressure drop of intermittent slug flow demonstrated in Figure 12c is intermittently fluctuated which appears at an inconspicuous period once in a while, manifesting signals with characteristics of low complexity and less information. As shown in Figure 12d,e, the differential pressure tendency is roughly consistent. The periodic period of signal also indicates that the flow pattern contains relatively low complexity and less information. Figure 12f represents differential pressure signal behavior of annular flow which is not that obvious. It can be noted that the features of annular flow signals are the most regular, indicating an overall low complexity and information. 

Beyond that, it should be noted that the low-frequency components of typical main-frequency differential pressure signals are in range of 0~5 Hz, and the global frequency components indicate that bubble flow has the most complex shape whereas annular flow is less than the other flow regime.

#### 3.3.2. Analysis of Flow Patterns

With a view to analyze the difference of flow pattern transition line when flow velocity changes, the SampEn of five typical flow patterns were extracted and analyzed. As mentioned before, the experiments were conducted with fixed water flow rate before increasing gas flow rate in each condition. Overall calculated results were shown in Figure 13. 

Figure 13 shows the SampEn of all typical flow patterns of gas-liquid two-phase flow in horizontal channel under heaving motion, especially in low scales, indicating that fluctuations of gas-liquid flow under slow flow rate are not that easily distinguished in low scales, and tend to be separable as the scale increases. It can be found that the SampEn of bubbly flow is greater than all other flow regimes, demonstrating the highest complexity and irregularity, which is in agreement with what is mentioned above. Flow characteristics reflected by features of SampEn shown in Figure 13 are specified as follows:

*Bubbly flow:* For bubbly flow, the SampEn increases with the increasing scale and then decreases abruptly at scale of 15. After that, it begins to grow in wave which is more obvious at lower liquid phase flow rate. As shown in Figure 3a, small bubbles randomly dispersed on the upside wall of the channel, which is similar with fluid distribution in stable channels. Most bubbles move forward with the main stream which is liquid phase. From the perspective of signal analysis theory, as scale increases, the frequency of multi-scale entropy decreased from high frequency to low frequency gradually in coarse-graining process. Due to the high complexity of bubble flow in the channel under heaving motion, the SampEn shows a fluctuant trend which is resemble with random signal. 

*Bead flow*: Variation tendency of multi-scale entropy for bead flow is similar to that for bubble flow, except for lower fluctuation amplitude after scale 14 as shown in Figure 11. It was observed from high-speed photography of Figure 3b that bubbles flow along with liquid phase in form of separate bead or small bubbles that stick together, which is different from flow phenomenon of stable channel under same work condition. Compared with Figure 3a, although small bubbles are accumulated on upper side of wall surface, bubbles in channel under heaving motion gradually swell into bead or pisolitic bubbles until forming a long enough gas column, which accordingly weakens the effect of vibration. 

The continuous generation of bubbles contributes to the sustainable growth of new energy spectra, which correspondingly increases multi-scale entropy with the increasing scale of bead flow. It can be noted that the magnitude of growth is more obvious at lower liquid flow rate indicating a more strongly separated and disturbed bubble motion in the channel. Beyond that, from the point of signal analysis theory, the degree of increasing multi-scale entropy of bead flow after scale 13 is lower and smoother than that of bubble flow, showing a flow condition of intermittent. Whereas, the motion of bead or pisolitic bubble is more biased toward random, contributing to a higher SampEn and complexity than fluctuant slug flow.

*Intermittent slug flow*: As shown in Figure 3c, compared with proto slug flow in channel under stable condition, bubbles of strip shape in channel under heaving motion are divided into small air parcels, each of which is surrounded by a mass of small bubbles. During the process of vibration, gas and liquid phase is distributed in a new state which where a gas column and water bomb appear alternately in the channel. These small bubbles are either suspended in the middle of channel or stuck to wall surface with the fluctuation of liquid layer, which also contains a large number of tiny bubbles. 

It was shown in Figure 13 that the SampEn of fluctuant slug flow rises smoothly with the increasing scale and then diminishes at a scale of 12. After that, it begins to increase undulately which is more obvious at a higher liquid phase flow rate. Compared with flow regime mentioned above, the random phenomenon of bubbles in fluctuant slug flow is not as obvious as that in bubble flow, which leads to a slight fluctuation of SampEn after scale of 12 especially at lower liquid flow rate. At the same time, regular alternation of gas and liquid phase causes a certain periodicity of differential pressure signal so that the SampEn is relatively low.

*Boiling wave flow*: Figure 3d,e displayed two major forms of stratified flow in horizontal channel under stability, which is the most widespread flow regime in gas-liquid two-phase flow. However, there is a significant difference in fluid distribution of the channel under heaving motion. The new flow regimes were defined as slug-wave flow and boiling wave flow, respectively. For slug-wave flow, liquid phase flow along with channel in form of wave. At the same time, under the effect of heaving motion, there are mounts of small bubbles attaching around liquid phase. For boiling-wave flow, gas phase is not only present in upper part of channel, but also in liquid gap to fill the entire pipe caused by irregular motion of the liquid phase. The entire flow pattern is similar to wavy flow, but with a more violent fluctuation of liquid interface ripple and lots of bubbles inside the liquid phase. 

As mentioned before, differential pressure signal of slug-wave flow is parallel with boiling flow, thus only multi-scale entropy of boiling flow was calculated as shown in Figure 13. It can be observed that SampEn rises gently with the increase of scale and deceases slightly at scale of 11, followed by a fluctuation within a narrow range. Although the flow behavior of boiling flow in channel is random, the magnitude of randomness is not as strong as that in bubble flow or bead flow. Therefore, the multi-scale of entropy is lower than other flow regime with bubbles. 

*Annular flow*: As can be seen from Figure 3f, a continuous gas column flows fast through the center of the channel, where liquid phase flows forward in the form of liquid film distributed around channel surface. As volume rate of gas phase increases, portions of liquid phase were entrapped in the formation of liquid filament at saw-tooth-shaped gas-liquid interface. Meanwhile, there is not much difference of flow regime in horizontal channel between stable condition and heaving motion, indicating that the effect of vibration on annular flow is not significant.

The multi-scale entropy of annular flow is stable with a light decline at scale 9, which is more obvious at a higher liquid phase flow rate. Owing to the inherent regularity and low flow complexity of annular flow signal, the SampEn is the minimum of all flow patterns mentioned before.

By comparison of Figure 13a,b, it can be observed that liquid phase flow rate has a certain effect on value of multi-scale entropy, but has no influence on variation trend of that for different flow patterns. Wave amplitude of SampEn for different flow regime was calculated as shown in Table 2.

As shown in Table 2, the influence of liquid phase flow rate on multi-scale entropy is more significant at a higher scale, which can reach 37% from bead flow to annular flow. Contrary to that, the multi-scale entropy of individual flow regime with different liquid phase flow rate is almost the same when the scale is less than 14. It can also be observed that the random motion of bubbles has an obvious influence on multi-scale entropy by comparing ΔSampEn of fluctuant slug flow and boiling wave flow, which is 10% higher than transition of other flow patterns either at high flow rate or low. Beyond that, the fluctuation amplitude at the higher scale of flow contains bubbles and is more violent than other flow regime as shown in Figure 13, thus is agreement well with what is mentioned above. For bubble flow, wave amplitude of multi-scale entropy is higher at medium liquid phase flow rate rather than the highest flow rate, which confirms the effect of bubble motion, corresponding that the increasing flow rate magnifies multi-scale entropy on the one hand, but also enlarges bubble size which reduces the randomness of bubble movement, thus reducing multi-scale entropy.

At the same time, there is nearly no effect of heaving motion on annular flow. Notwithstanding, it should be pointed out that according to extensive experiments conducted in the present study, the liquid film thickness at the upper and bottom of the channel wall surface is basically identical when liquid flow rate is relatively lower, but when the liquid flow rate is great, most of liquid phase is still attached to bottom of channel wall which contributes to a thicker liquid film. This is also the reason why there is higher multi-scale entropy at higher liquid flow rate of annular flow. 

To sum up, liquid phase flow rate has some effect on the value of multi-scale entropy by affecting distribution of gas-liquid two-phase flow in horizontal channel under heaving motion but has little impact on the boundary of multi-scale entropy for different flow regime. It also shows that multi-scale entropy can not only reflect the unique changing trend of various flow patterns at different scales macroscopically, but also reveal dynamic characteristics and evolution process of gas-liquid two-phase flow in channel under heaving motion from microscopic perspective.

#### 3.3.3. Analysis of Flow Patterns with Different Vibration Parameters 

As discussed above, transition boundary of multi-scale entropy for flow patterns was affected by the change of liquid phase flow velocity. Therefore, it is possible for vibration parameters to have a similar influence on gas-liquid two-phase flow in channel under heaving motion. Since there is little influence of heaving motion on bubble flow and annular flow in horizontal channel, only three typical flow patterns with different vibration parameters including vibration frequency and amplitude were calculated, results were shown as Figure 14.

As shown in Figure 14, as vibration amplitude and frequency rise, SampEn of each flow regime at small scale slightly increase, revealing that the instability is aggravated by the increase of vibration parameters. After the scale of 10, the multi-scale entropy of bead flow and fluctuant slug flow appear to have a more obvious fluctuation with higher vibration amplitude and frequency, whereas the other flow pattern exhibits no distinct difference except for some points with a slightly higher value. Meanwhile, by comparing Figure 14a,b, it can be observed that vibration frequency has a greater impact on multi-scale entropy value of gas-liquid two-phase flow in the channel under heaving motion than vibration amplitude. The reasons accounted for phenomena mentioned above can be illustrated as follows:

Generally, factors influencing pressure drop of gas-liquid two-phase flow in channel under steady condition mainly include wall properties and Re number which is affected by channel geometry, fluid velocity and properties. Besides that, convection effect of fluid in channel under heaving motion is affected by vibration frequency, vibration amplitude, fluid viscosity and other parameters. Thus, instantaneous pulsating vortices aroused from heaving motion will occur in a thin area of channel wall, thus inducing tremendous pulse velocities which will transfer vibration amplitude within boundary layer to the interior of fluid, thereby affecting flow field structure. The greater the vibration amplitude and frequency, the more significant the influence on original flow field structure, contributing to a more disordered flow condition which represents a greater extent of deviation of streamlines from the original direction. In the case of small vibration amplitude and low frequency, vortices of large scale become small gradually, and then those small vortices will break up and flow along with main stream contrary to those under large vibration amplitude and high frequency where large vortices break up directly and then flow into main stream. As a result, fluctuation of SampEn is more obvious under high vibration amplitude and frequency. Compared with vibration amplitude, bubbles are more susceptible to vibration frequency, which has a greater impact on multi-scale entropy as mentioned above. Thus, the fluctuation amplitude of flow contains bubbles that are more pronounced at higher vibration frequency than that at higher vibration amplitude, and boiling wave flow seems more stable with variation of vibration parameters, which is also consistent with the influence of liquid phase flow rate.

### 3.4. Flow Pattern Separation

As already noted, the SampEn at around scale 11 showed the most distinctive response when flow pattern and flow rate change, therefore it was adopted as an index for flow pattern separation. Meanwhile, as indicated from Figure 13 and Figure 14, multi-scale entropy increases from scale 1 to 25. Thus, as another index, mean SampEn of the instantaneous pressure drop signals of six flow patterns under different vibration conditions was employed. The results are shown in Figure 15, revealing the variation of flow pattern with fluid flow rate. 

As displayed in Figure 15, 40% of points were in a *y* = *x* line with the SampEn of scale 11, meaning an equivalent between mean SampEn and SampEn of scale 11. The reason for this can be attributed to monotonic increase of SampEn versus scale as shown in Figure 15. Other data points deviated from the line can be ascribed to variation of vibration parameters and flow pattern transition such as from fluctuant slug flow to boiling wave flow. Mean SampEn of bubble flow and bead flow are almost located in *y* = *x* line at top right corner of Figure 15. By increasing flow velocity rate, fluctuant slug flow wave flow occurs and begins deviating from the line resulted from nonlinear variation of multi-scale entropy versus scales as shown in Figure 13. Meanwhile, mean SampEn of flow regime also deviates greatly from the line by increasing vibration parameters as illustrated in Figure 15. Further increasing fluid flow rate, it can be found that data points of mean SampEn turn round to the *y* = *x* line, showing a regularity of annular flow.

Moreover, a confusing issue occurs that the SampEn of boiling wave flow and slug-wave flow is mixed together, causing a comparatively difficulty in separating these two flow patterns. Combined with flow characteristics in Figure 3d,e, it can be analyzed that although there are some small bubbles showing random fluctuations at gas-liquid interface of slug-wave flow, periodic waves still exist that result in long range correlation which is similar to the boiling wave flow. Thus, mean value of multi-scale entropy shows nearly ballpark response of the combination of small bubbles and waves contained both in slug-wave flow and boiling wave flow.

The characterization of Figure 15 indicates not only an effective separation of flow patterns, but also the progress of flow regime transition from bubble flow to annular flow and influence of vibration parameters on distribution of gas-liquid two-phase flow in horizontal channel under heaving motion. The only matter is the mixing part of mean SampEn for slug-wave flow and boiling wave flow at scale 11. Therefore, 100 sets of differential pressure fluctuation signals of six typical flow patterns were collected, and the rate of multi-scale entropy of that was calculated by method of the least-squares. 

The selection of characteristic parameters can directly determine the reaction of flow regime and identification. Figure 16a is a flow pattern figure of fluid flow under heaving motion, which was formed by flow conditions and entropy rate using 25 scales. Each gas-liquid velocity combination represents a flow condition. In this section, 100 different gas-liquid flow rates are selected, which represents 100 flow conditions. Meanwhile, as illustrated in Figure 4a, the transition boundary of flow regime under steady state obtained from our experimental results can also be applied to the traditional flow map provided by Ran et al. [29]. Therefore, to verify the prediction accuracy of the new entropy- related method, a flow pattern figure of fluid flow under steady state was also shown in Figure 16b.

From the distribution of sample in the plane, it is divided into five parts from Y1 to Y5 as shown in Figure 16a. At the same time, the data illustrated in Figure 16b verified the correctness of our method in predicting flow regime transition line in horizontal channel under steady state. In Figure 16a, multi-scale entropy rate of bubble flow ranges from 0.16 to 0.185 located in top left corner as shown in Y1, the variation range of entropy rate for bead flow is from 0.122 to 0.159 with a location between Y1 and Y3 which represents fluctuant slug-flow with a rangeability of entropy rate from 0.095 to 0.122. As for annular flow situated in lower right corner, multi-scale entropy rate of it is below 0.055 defined as Y5. However, it should be noted that the transition boundary of slug-wave flow and boiling wave flow is difficult to distinguish with a single feature as shown in Y4; there is only slight difference between two flow patterns, thus two-dimensional joint distribution of convective-type recognition needs to be further analyzed. By and large, multi-scale entropy rate can accurately identify flow patterns with a recognition rate of 94.56%, indicating that the method adopted in present study is reliable in identifying gas-liquid two-phase flow condition in horizontal channel under heaving motion. 

## 4. Analysis of Flow Pattern Based on Multi-Scale Marginal Spectrum Entropy

As it can be seen from Figure 16, it should be noted that there is only slight difference between the transition boundary of slug-wave flow and boiling wave flow. Consequently, it is difficult to distinguish these two kinds of flow regime with a single feature by method of multi-scale entropy. Therefore, another method of characterizing gas-liquid two-phase flow pattern in horizontal channel under heaving motion based on the use of multi-scale marginal spectrum entropy is discussed as follows:

### 4.1. Theory Basis

Multi-scale marginal spectrum entropy is based on HHT (Hilbert-Huang Transform) and marginal spectrum. After decomposing coarse-grained signal by EMD (Empirical Mode Decomposition), a group of finite IMF (Intrinsic Mode Function) components expressing signal will be obtained. By method of Hilbert transform, the spectrum is acquired by searching for the instantaneous amplitude and frequency. Combining the definition of information entropy, algorithm of multi-scale marginal spectrum entropy is defined as follows [34,35]:

1. EMD decomposition is performed after coarsening signal which is presented by finite IMF components:
(16)$$x\left(t\right)=\sum_{i=1}^{N-m}c_{i}\left(t\right)+r\left(t\right)$$
where *c_i_*(*τ*) represents *N* − *m* IMF model components and *r*(*τ*) is residual term.

2. The Hilbert transform for each IMF component is obtained as
(17)$$H\left[c\left(\tau \right)\right]=\frac{1}{\pi }\int_{-\infty }^{+\infty }\frac{c\left(\tau \right)}{t-\tau }d\tau$$


3. Searching for instantaneous amplitude and frequency. The amplitude function can be represented as $a\left(t\right)={\left(c^{2}\left(t\right)+H^{2}c\left(t\right)\right)}^{1/2}$, where $c\left(t\right)=a\left(t\right)\cos\varphi \left(t\right)$. The instantaneous frequency of $c\left(t\right)$ can be obtained by $f^{i}\left(t\right)=\frac{1}{2\pi }\cdot \frac{d}{dt}\left(\varphi \left(t\right)\right)$.

4. Defining Hilbert spectrum as $H\left(f^{i},t\right)=Re\sum_{k=1}^{N-m}a_{k}\left(t\right)e^{j2\pi \int f_{k}^{i}\left(t\right)dt}$.

5. Marginal spectrum is obtained by $h\left(f^{i}\right)=\int_{0}^{T}H\left(f^{i},t\right)dt$. 

### 4.2. Flow Pattern Separation

As can be seen from Figure 17, a two-dimensional distribution map of calculated entropy rate with respect of mean entropy for 100 set of various flow conditions is provided. Entropy rate is defined as the rate of entropy increase before pressure-differential signal of each flow regime mutation, where the slope is obtained by method of least-squares.

The whole flow regime map is divided into six parts by five lines: *y_n_* = *k_n_x* + *b_n_*. Based on method of least square fitting, the slope *k_n_* of each flow regime can be obtained. According to the actual distribution of flow profile entropy value, *b_n_* can be obtained. Therefore, the boundaries of six flow regimes provided in Figure 15 are: *y*_1_ = 0.01177*x* − 0.06, *y*_2_ = 0.023*x* + 0.02, *y*_3_ = 0.0049*x*, *y*_4_ = 0.007*x* − 0.003, *y*_5_ = 0.0301*x* − 0.02.

Obviously, compared with a single feature, for recognition performance of flow regime in the same working condition, the two-dimensional joint distribution of convective-type recognition is more accurate. In particular for transition line of slug-wave flow and boiling wave flow.

## 5. Error Analysis

Concerning the applicability of flow pattern studies and their relationship with multi-scale entropy, there exist some common questions. Due to the limited length of this paper, there are some reasons for error analysis for the identification of flow regime: (1) the variation range of transition flow parameter is large and overlaps that of the adjacent flow regime; (2) the number of training samples is insufficient; (3) the cumulative error of experimental loop caused by contamination is allowed. Owing to the advantages of fast identification and high recognition rate included in the proposed identification model, it can be applied to on-line identification of engineering flow regime. 

## 6. Conclusions

A new method of characterizing gas-liquid two-phase flow pattern in horizontal channel under heaving motion is presented based on the use of multi-scale entropy to analyze differential pressure signal of various flow regime. The measurement time-series from each flow regime are treated as a multivariate time-series and then processed with multi-scale entropy to identify flow characteristics of gas-liquid flow. In the present study, multi-scale entropy of 100 flow conditions classified to six typical patterns was calculated and the evolution law of various flow regimes was determined by comparison. From the work reported herein, the following conclusions were drawn:

(1) Vibration parameters have a significant effect on gas-liquid two-phase flow behavior in horizontal channel under heaving motion, whereas little influence on transition lines. Six typical flow regimes appeared in channel are bubble flow, bead flow, annular flow, fluctuant slug flow, slug-wave flow and boiling wave flow. 

(2) Multi-scale entropy is a frequency-domain entropy analysis method with a definite physical meaning. It can not only qualitatively distinguish different flow patterns from macroscopic ones, but also can quantitatively present the sensitivity of dynamics characteristics of a particular flow regime to scale in detail and indicate the flow evolution rule. 

(3) For dynamic working conditions adopted in the present experiment, magnitude of multi-scale entropy value has been influenced by vibration parameters slightly, but is unacted on distinguishing flow regime.

(4) Compared to the commonly used BP neural-network, SampEn method can accurately identify flow patterns with a recognition rate of 94.56%. It is more suitable for small-sample calculations owing to its fast calculation and high recognition rate. 

(5) For recognition performance of transitional flow regime, the two-dimensional joint distribution of convective-type recognition is more accurate, which provides a new direction for pattern identification.

## Figures and Tables

**Figure 1 entropy-21-00667-f001:**
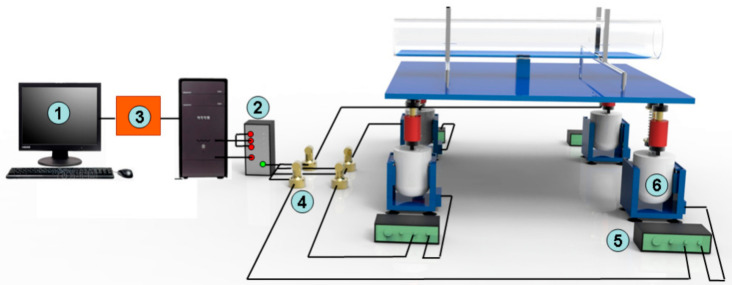
Configuration of vibration control system (1—industrial control computer; 2—multi-channel parallel collection; 3—asynchronous collaborative control software; 4—capacitance transducer; 5—digital switching power amplifier; 6—electromagnetism oscillator).

**Figure 2 entropy-21-00667-f002:**
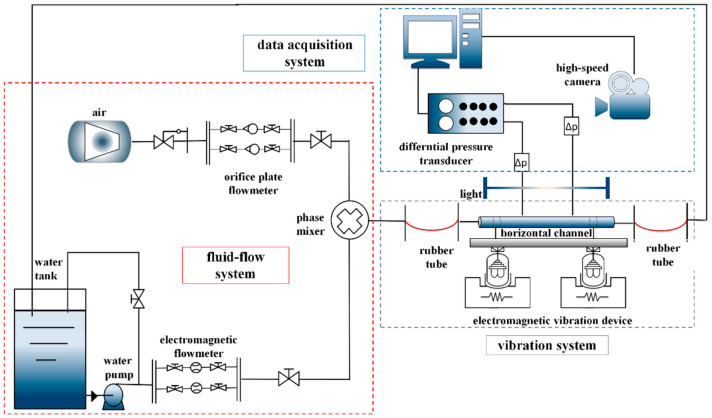
Gas-liquid two-phase flow experimental system.

**Figure 3 entropy-21-00667-f003:**
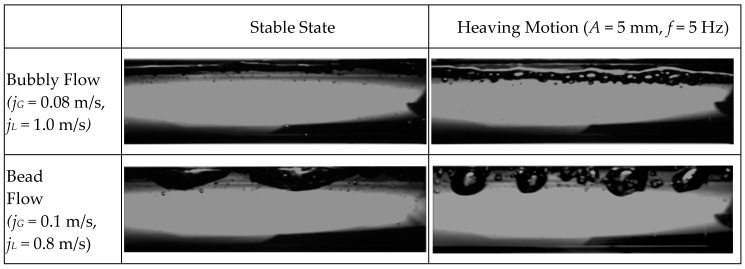
High-speed photograph of flow pattern under steady state and heaving motion.

**Figure 4 entropy-21-00667-f004:**
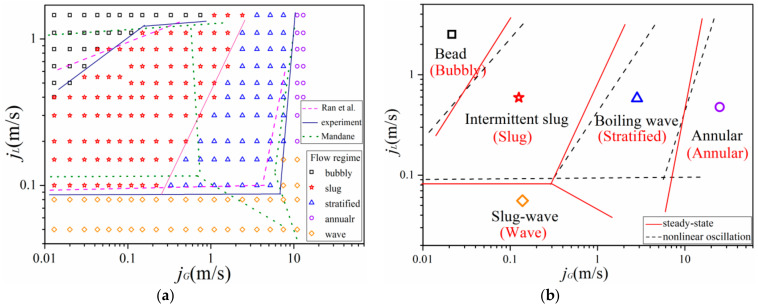
Comparison of flow regime map for horizontal flow under (**a**) steady state with traditional flow maps and (**b**) nonlinear oscillation with experimental results (*A* = 5 mm, *f* = 5 Hz).

**Figure 5 entropy-21-00667-f005:**
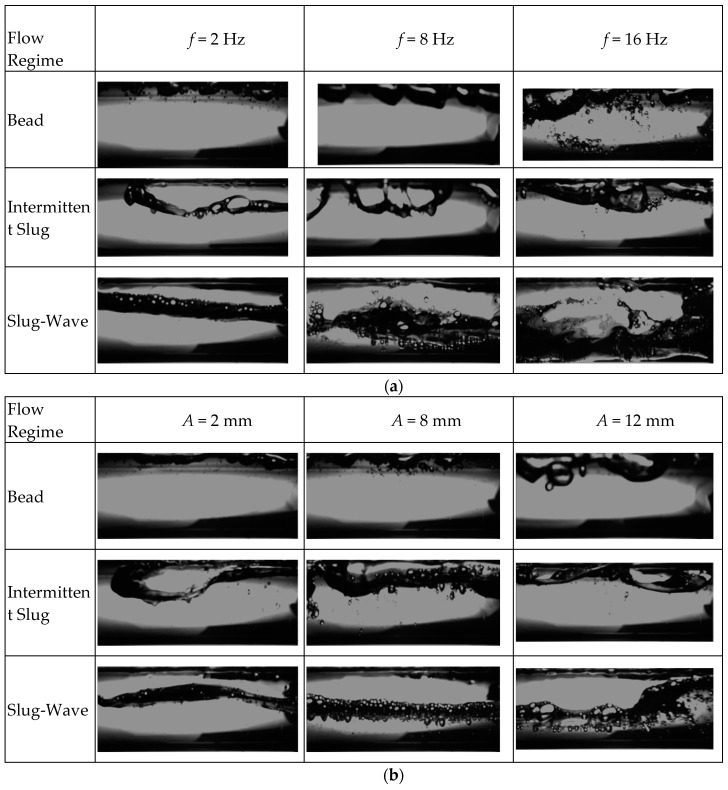
(**a**) High-speed photograph of flow pattern with various vibration frequency (*A* = 5 mm); (**b**) High-speed photograph of flow pattern with various vibration amplitude (*f* = 5 Hz).

**Figure 6 entropy-21-00667-f006:**
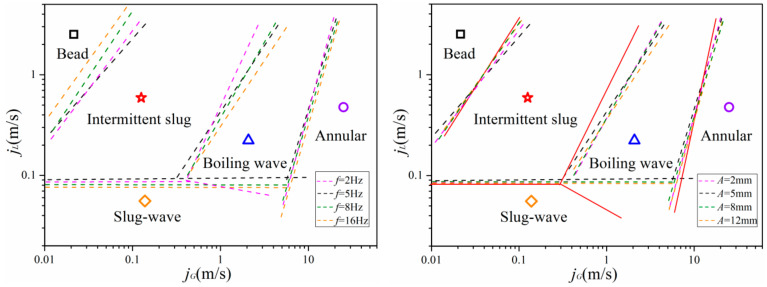
Flow pattern transition line with various vibration parameters.

**Figure 7 entropy-21-00667-f007:**
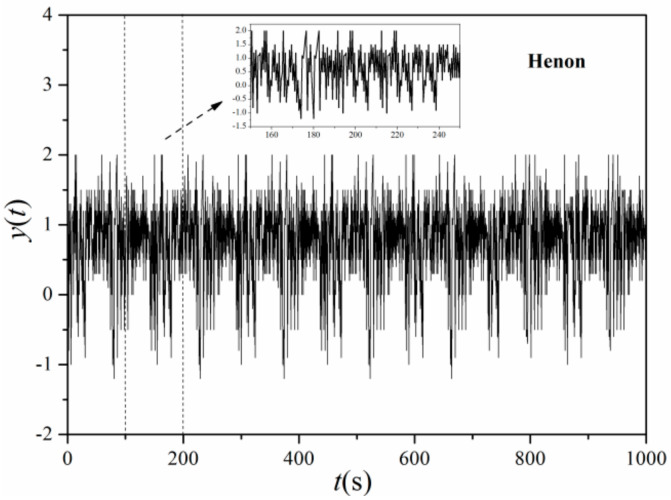
Signals produced by Henon system.

**Figure 8 entropy-21-00667-f008:**
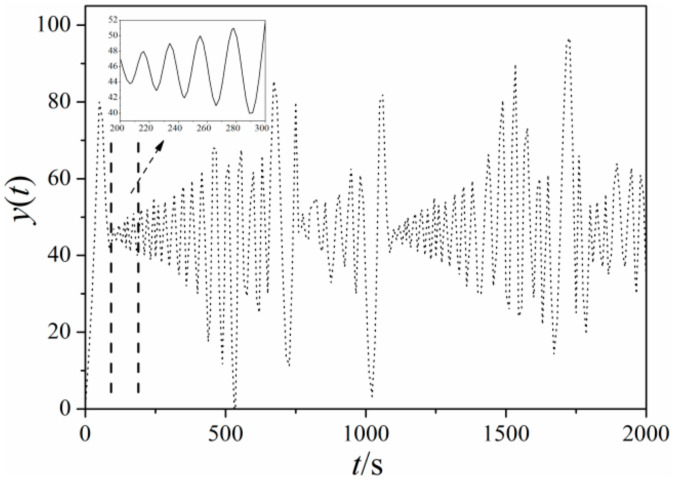
Signals produced by Lorenz system.

**Figure 9 entropy-21-00667-f009:**
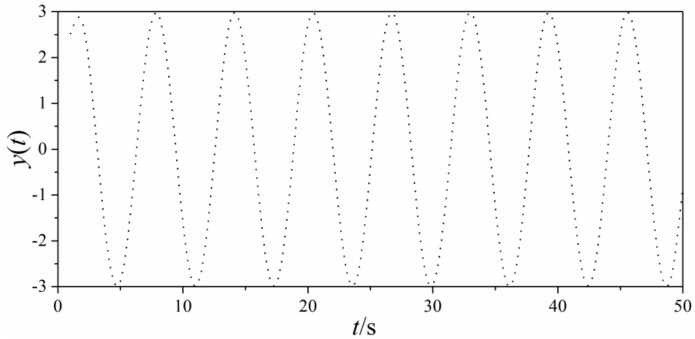
Sine signals produced by Equation (15).

**Figure 10 entropy-21-00667-f010:**
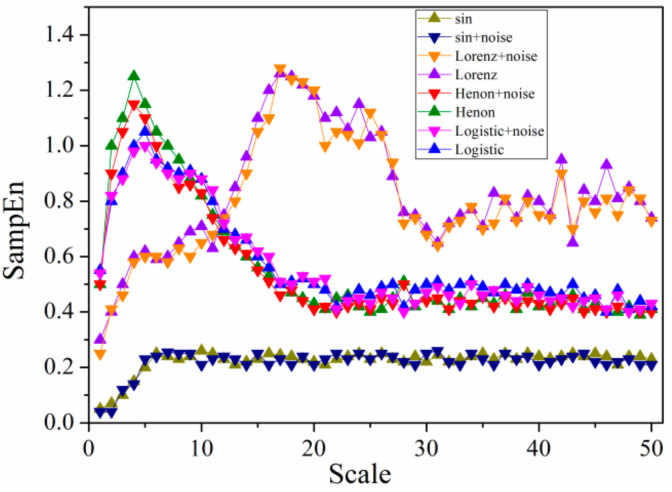
Values of multi-scale entropy with scale factors for eight typical signals.

**Figure 11 entropy-21-00667-f011:**
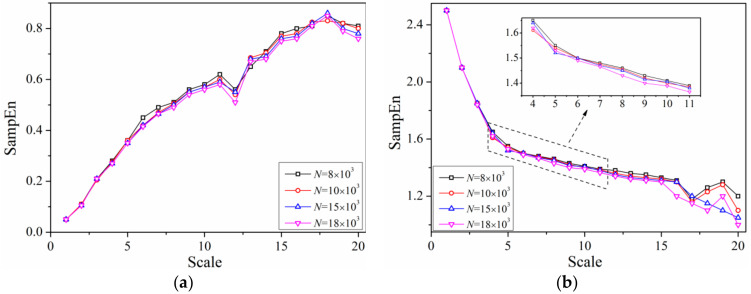
Multi-scale entropy of slug flow with different sequence lengths. (**a**) Slug flow (*j_G_* = 0.25 m/s, *j_L_* = 0.3 m/s) (**b**) Gaussian.

**Figure 12 entropy-21-00667-f012:**
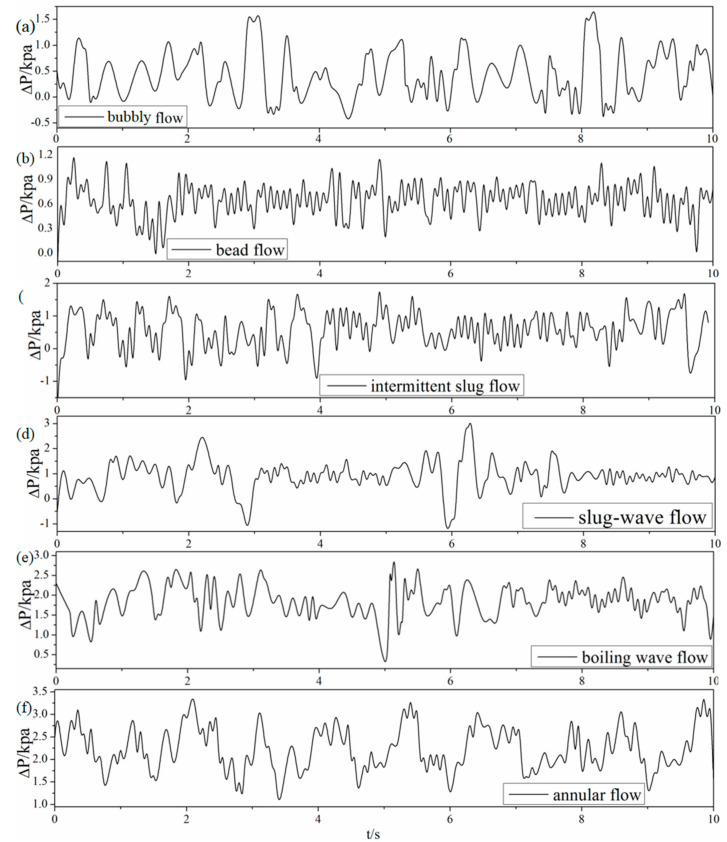
Differential pressure fluctuation signals of six flow regimes.

**Figure 13 entropy-21-00667-f013:**
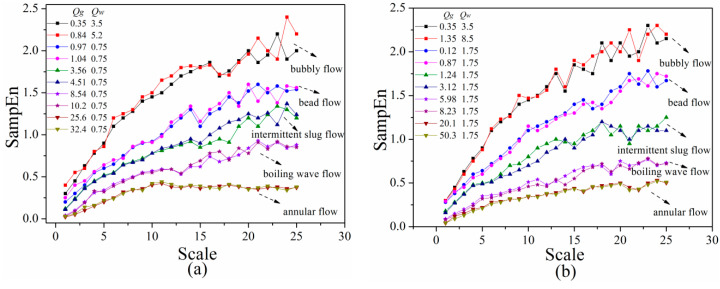
Multi-scale entropy of pressure fluctuation signals for (**a**) *Q_w_* = 0.75 m^3^/h and (**b**) *Q_w_* = 1.75 m^3^/h with five typical flow regimes (*A* = 5 mm, *f* = 5 Hz).

**Figure 14 entropy-21-00667-f014:**
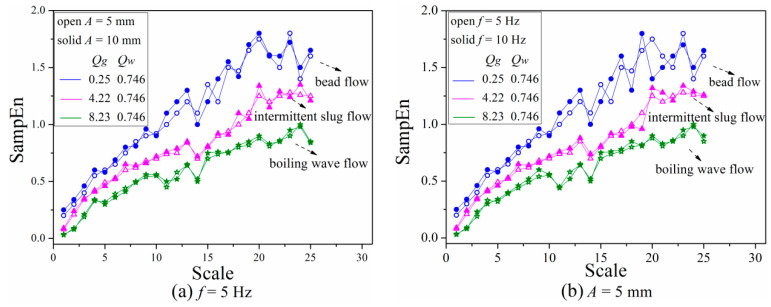
Multi-scale entropy of three typical flow regimes with different vibration parameters. (**a**) Vibration amplitude; (**b**) Vibration frequency.

**Figure 15 entropy-21-00667-f015:**
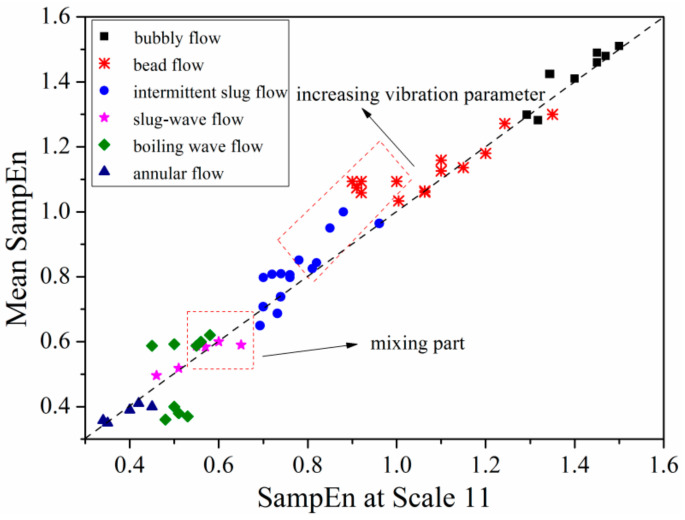
Flow pattern separation with SampEn at scale 11 and the average SampEn of all scales.

**Figure 16 entropy-21-00667-f016:**
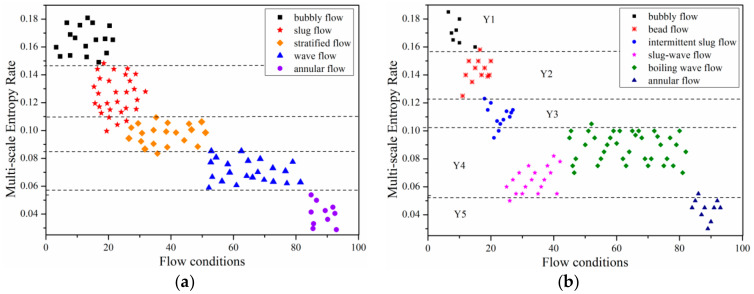
Multi-scale entropy rate of six typical flow regimes for (**a**) steady state; (**b**) heaving motion (*A* = 5 mm, *f* = 5 Hz).

**Figure 17 entropy-21-00667-f017:**
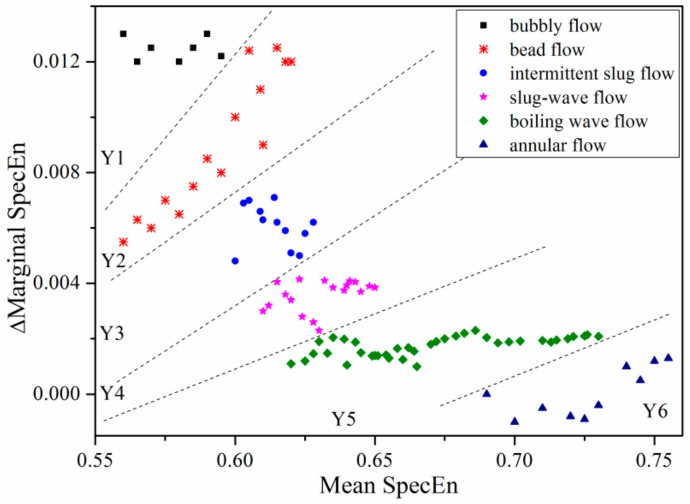
Multi-scale marginal spectrum entropy generation rate with respect to mean spectral entropy of six typical flow regimes (*A* = 5 mm, *f* = 5 Hz).

**Table 1 entropy-21-00667-t001:** Parameters of vibration platform.

Parameter	ETS MPA101A/L215M
Rated thrust peak	1964.33 n
Rated peak ground acceleration	100.034 g
Rated peak velocity	2 m/s
Rated displacement peak	25.4 mm
Vibration frequency	2.00~4500 Hz
Maximum input voltage of amplifier	10 V
Equivalent mass of moving parts	2 kg
Total load	2 kg

**Table 2 entropy-21-00667-t002:** Wave amplitude of SampEn for different flow regime.

ΔSampEn		Bead Flow	Fluctuant-Slug Flow	Boiling-Wave Flow	Annular Flow
scale 0–14	*Q_w_* = 0.75	0.780	0.610	0.412	0.254
	*Q_w_* = 1.75	0.833	0.620	0.363	0.264
Scale 15–25	*Q_w_* = 0.75	1.430	1.129	0.806	0.373
	*Q_w_* = 1.75	1.545	1.088	0.691	0.464

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
