# Peer review of "Flow Regime Recognition and Dynamic Characteristics Analysis of Air-Water Flow in Horizontal Channel under Nonlinear Oscillation Based on Multi-Scale Entropy"

_entropy, 2019, doi:10.3390/e21070667_

Round 1

Reviewer 1 Report

p.p1 {margin: 0.0px 0.0px 0.0px 0.0px; font: 12.0px 'Helvetica Neue'} li.li1 {margin: 0.0px 0.0px 0.0px 0.0px; font: 12.0px 'Helvetica Neue'} ul.ul1 {list-style-type: hyphen}

Sun et al. report an experimental study on two-phase flow regime identification in a horizontal pipe under vibration conditions using multi scale entropy techniques. The paper is interesting, written in good English with a solid theoretical background. It is worth publishing in Entropy. I only have minor comments:

Line 161: space between “of” and “±0.2%”. Same for line 163

Please space your numbers and units e.g. 2bar (should be 2 bar) in line 161, 0.03mm in line 163 and others all over the manuscript. 

Space your 

Fig. 9 caption, change to “Sine signals produced by equation 15”

What does the horizontal axis mean in Fig. 16? Flow conditions is ambiguous

What is the physical meaning of your “Scales” in the horizontal axes of your plots Fig. 10, 11, 13, 14? It is currently not defined, please kindly do so by way of a table or paragraph. If it is what is given in Table 2, it is not clear enough. 

So your SampEn in Eqn 10 that is used to calculate the sample entropy is based on the instanteneous pressure measurements? This was not explicitly stated in the manuscript. Please clarify.

How do you relate the flow regime demarcation in Fig. 17 with a typical flow regime map e.g. Taitel and Dukler 1976, for horizontal flow? Please comment. 

Author Response

Comment 1. Line 161: space between “of” and “±0.2%”. Same for line 163

Please space your numbers and units e.g. 2bar (should be 2 bar) in line 161, 0.03mm in line 163 and others all over the manuscript. 

Answer: Thank you very much for the reviewers correction. Space between numbers and units all over the paper has been added.

Comment 2. Space your Fig. 9 caption, change to “Sine signals produced by equation 15”

Answer: Thank you very much for the reviewers comment. The caption of Fig. 9 has been modified to “Sine signals produced by equation 15”, and space has been added.

Comment 3. What does the horizontal axis mean in Fig. 16? Flow conditions is ambiguous

Answer: Thank you very much for the reviewers comment. Each flow condition represents a combination of gas-liquid velocity. In this section, there are 100 different gas-liquid flow rates selected. Therefore, Fig. 16 includes 100 data. For better understanding, corresponding statement has been added in section 3.4.

Comment 4. What is the physical meaning of your “Scales” in the horizontal axes of your plots Fig. 10, 11, 13, 14? It is currently not defined, please kindly do so by way of a table or paragraph. If it is what is given in Table 2, it is not clear enough. 

Answer: Thank you very much for the reviewers comment. Firstly, the definition of scale has been defined in section 3.1:With the set scale of 1, the reconstruction is the original time series {u(i):i=1, 2,, N}. For a τ-scale, the time-series is coarse grained as {yτ(j)=1, 2,., N/τ}. Actually, the scale factor is a coefficient in a scale function, which represents the low frequency characteristics of the signal. According to previous literature [1], multi-scale is one yardstick to signal a scale to repeat folding.

[1] Zheng, G.B.; Jin, N.D. Multi-scale entropy and dynamic characteristics of two-phase flow patterns. Acta Physca Sinnica, 2009, 58: 4485-4492.

Comment 5. So your SampEn in Eqn 10 that is used to calculate the sample entropy is based on the instanteneous pressure measurements? This was not explicitly stated in the manuscript. Please clarify.

Answer: Thank you very much for the reviewers comment. We agree with the  reviewers comment that SampEn in Eqn 10 used to calculate the sample entropy is based on the instantaneous pressure measurements, which has been clarified in section 3.4.

Comment 6. How do you relate the flow regime demarcation in Fig. 17 with a typical flow regime map e.g. T aitel and Dukler 1976, for horizontal flow? Please comment. 

Answer: Thank you very much for the reviewers comment. As shown in Fig.4, a comparison of flow regime map under stable condition between our experimental results and traditional flow map obtained by Ran et al. [1] and Mandhane [2] has been illustrated. Beyond that, a flow regime transition boundary map of fluid flow in horizontal channel under heaving motion is also shown in Fig. 4(b). By comparing Fig. 4(a) and (b), it can be found that though nonlinear oscillation has a significant influence on the distribution of gas-liquid two-phase in channel, the transition boundary of different flow regime under nonlinear oscillation can also be applied to fluid flow in horizontal channel under steady state. Therefore, the flow regime demarcation in Fig. 17 can also be used in other typical flow regime map for horizontal flow with similar working conditions.

1. Ran, K.; Kim, S., Bajorek, S., Tien, K. C. Hoxie, Effects of pipe size on horizontal two-phase flow: Flow regimes, pressure drop, two-phase flow parameters, and drift-flux analysis. Experimental Thermal and Fluid Science,2018,96: 75-89.

2. Mandhane, J.M., Gregory, G.A., Aziz, K. A flow pattern map for gas-liquid flow in horizontal pipes, International Journal of Multiphase Flow, 1974, 1: 537-553.

Reviewer 2 Report

Dear authors,

this paper is an interesting study about the flow pattern recognition by using an innovative method based on multi-scale entropy.

The topic is up-to-date and the paper is well written and organized in all its sections. I suggest an English double-check for sporadic typos.

However, there are some major issues that need to be addressed before considering this paper suitable for publication. Specifically:

1) It is surprising not to find any reference in the introduction related to the traditional well-known flow pattern maps for gas-liquid flow, such as those of Taitel and Duckler, or Baker, etc. Please add these methods to the state of the art.

2) Please improve the quality of the presentation of your methods and results. Particularly, the operating temperature used for your experiments (ambient temperature?) is not provided in the manuscript. Then, it is not clear whether Figures 13, 16 and 17 refer to no motion of the tube (f=0 Hz) or to heaving motion. Please clarify these points.

Moreover, try to make all your axes style uniform (for instance the units of measurement are sometimes in brakets and sometimes after the / symbol).

3) Although the  method proposed by the authors is interesting and innovative, it is quite difficult to use it on a real test bench. In other words, it is not clear to the reader which is the real advantage of this method if compared to the conventional ones, that are much easier to implement. I therefore suggest to compare the prediction accuracy of the new entropy-related method to that of traditional flow pattern maps (Taitel and Dukler, Baker,...) when used with your experimental observations.

Author Response

Comment 1.  It is surprising not to find any reference in the introduction related to the traditional well-known flow pattern maps for gas-liquid flow, such as those of Taitel and Duckler, or Baker, etc. Please add these methods to the state of the art.

Answer: Thank you very much for your correction. A comparison of flow regime map under stable condition between our experimental results and traditional flow map obtained by Ran et al. [1] and Mandhane [2] has been added in Fig.4(a). Beyond that, a flow regime transition boundary map of fluid flow in horizontal channel under heaving motion is also added in Fig. 4(b).

1. Ran, K.; Kim, S., Bajorek, S., Tien, K. C. Hoxie, Effects of pipe size on horizontal two-phase flow: Flow regimes, pressure drop, two-phase flow parameters, and drift-flux analysis. Experimental Thermal and Fluid Science,2018,96: 75-89.

2. Mandhane, J.M., Gregory, G.A., Aziz, K. A flow pattern map for gas-liquid flow in horizontal pipes, International Journal of Multiphase Flow, 1974, 1: 537-553.

Comment 2.  Please improve the quality of the presentation of your methods and results. Particularly, the operating temperature used for your experiments (ambient temperature?) is not provided in the manuscript. Then, it is not clear whether Figures 13, 16 and 17 refer to no motion of the tube (f=0 Hz) or to heaving motion. Please clarify these points. Moreover, try to make all your axes style uniform (for instance the units of measurement are sometimes in brakets and sometimes after the / symbol).

Answer: Thank you very much for the for the reviewers comment. Firstly, the presentation of your methods and results has been improved carefully. The operating temperature used for our experiments is ambient temperature, which has been added in section 2.3. Secondly, Figs. 13, 16 and 17 refer to heaving motion,which has been clarified in caption of figures. Lastly, style of all axes in this paper has been uniformed.

Comment 3.Although the  method proposed by the authors is interesting and innovative, it is quite difficult to use it on a real test bench. In other words, it is not clear to the reader which is the real advantage of this method if compared to the conventional ones, that are much easier to implement. I therefore suggest to compare the prediction accuracy of the new entropy-related method to that of traditional flow pattern maps (Taitel and Dukler, Baker,...) when used with your experimental observations.

Answer: Thank you very much for the for the reviewers comment. Since the method needs pressure drop data of different flow regime, it seems not possible to apply the new entropy-related method to that of traditional flow pattern maps. However, as illustrated in Fig. 4(a), the transition boundary of flow regime under steady state obtained from our experimental results can also be applied to traditional flow map provided by Ran et al. [1]. Therefore,we think it is reasonable to verify the prediction accuracy of the new entropy- related method by applied the method to data of pressure drop obtained from our experiment under steady state. The results were shown in Fig. 16(b). we hope the comparison can verify the prediction accuracy of the new method.

1. Ran, K.; Kim, S., Bajorek, S., Tien, K. C. Hoxie, Effects of pipe size on horizontal two-phase flow: Flow regimes, pressure drop, two-phase flow parameters, and drift-flux analysis. Experimental Thermal and Fluid Science, 2018, 96: 75-89.

Reviewer 3 Report

There are typos and grammatical errors. Please re-read complete article to fix. Overall it is an well presented article. For Introduction the authors may visit following recent article which presents numerical simulation of two-phase flow. I recommend for publication upon minor revision.

Akand W. Islam, Alexander Sun, Kamy Sepehrnoori, "An Efficient Computational Scheme for Two-Phase Steam Condensation in the Presence of CO2 for Wellbore and Long-Distance Flow", ChemEngineering, 2019, 3, 4.

Author Response

Answer: Thank you very much for the for the reviewers comment. The manuscript has been re-written and re-organized carefully, typos and grammatical errors have been modified. The article "An Efficient Computational Scheme for Two-Phase Steam Condensation in the Presence of CO2 for Wellbore and Long-Distance Flow", ChemEngineering, 2019, 3, 4. has been added in section of Introduction.

Round 2

Reviewer 1 Report

The authors have satisfactorily responded to my queries, and I have no objections to the manuscript being accepted. Only a few minor observations:

Table 1: why did the authors move the units from the ''parameter'' column to the ''ETS'' column with the numbers? It is preferable as it was before.

Just before Fig. 8, correct the spelling of ''Runde-Kutta'' to ''Runge-Kutta''. 

Other adjustments are purely cosmetic and will be taken care of at the proofing and production stages. 

Thank you for your contribution and all the very best. 

Reviewer 2 Report

No more remarks. Congratulations for your work

Entropy EISSN 1099-4300 Published by MDPI AG, Basel, Switzerland RSS E-Mail Table of Contents Alert
Back to Top